# How breathing disrupts vision: hyperventilation-induced hypocapnia impairs oculomotor responses in resting humans

Yusei Yoshimura[1] , Tomoka Sagawa[1], Seiji Ono[1,2], Takeshi Nishiyasu[1,2] and Naoto Fujii[1,2]

[1]*Institute of Health and Sport Sciences, University of Tsukuba, Tsukuba, Japan*
[2]*Advanced Research Initiative for Human High Performance (ARIHHP), University of Tsukuba, Tsukuba, Japan*

Handling Editors: Richard Carson & Frank Powell

The peer review history is available in the Supporting Information section of this article (https://doi.org/10.1113/JP289870#support-information-section).

**Abstract figure legend** Hypocapnic hyperventilation reduced end-tidal carbon dioxide partial pressure and middle cerebral artery mean velocity, and impaired oculomotor response by modulating visual fixation and anti-saccadic

**Yusei Yoshimura**, received his doctorate in health and sport sciences from the University of Tsukuba in 2024, where he was mentored by Professor Seiji Ono, and is currently a junior assistant professor at the same university. His research focuses on visual cognition, including attention and perception, in both sports contexts and everyday life. He investigates these processes using eye-tracking and psychophysical methods.

control. Hyperventilation itself also impaired anti-saccadic control. These oculomotor impairments, mediated by hypocapnic hyperventilation, may increase the risk of injury or death during tasks that require precise visuomotor coordination.

**Abstract** Eye movements are precisely controlled by the brain to acquire clear and stable visual information, and eye movement measurements are also used as neurophysiological biomarkers. Hyperventilation, which reduces arterial carbon dioxide partial pressure (hypocapnia) and cerebral perfusion, can be triggered by environmental or psychological stress or by chronic disease conditions. Here, we hypothesized that hyperventilation-induced hypocapnia would impair oculomotor responses in resting humans. Thirteen healthy young adults (eight females) performed a free-viewing task and an anti-saccade task under three breathing conditions: spontaneous breathing, voluntary hypocapnic hyperventilation and voluntary normocapnic hyperventilation. Eye movements were recorded using video-based eye tracking, whilst end-tidal carbon dioxide partial pressure and middle cerebral artery mean blood velocity were continuously monitored via a metabolic cart and transcranial Doppler ultrasound, respectively. Hypocapnic hyperventilation reduced end-tidal carbon dioxide partial pressure to ∼20 mmHg, with a concurrent $24 \pm 10$ cm/s reduction in middle cerebral artery blood flow (both $P < 0.001$). Hypocapnic hyperventilation also reduced the number of fixations and saccades, and scanpath length, whereas it increased fixation duration in the free-viewing task (all $P < 0.01$). The aforementioned responses mediated by hypocapnic hyperventilation were not observed under spontaneous breathing or normocapnic hyperventilation conditions (all $P > 0.11$). In the anti-saccade task, both normocapnic and hypocapnic hyperventilation prolonged latency (both $P < 0.01$), with hypocapnic hyperventilation exhibiting greater impairment ($P < 0.001$). We show that hyperventilation-mediated hypocapnia impairs oculomotor responses by attenuating visual fixation and saccadic control in resting humans. Also, hyperventilation *per se* independently of hypocapnia impairs saccadic control.

(Received 5 August 2025; accepted after revision 2 January 2026; first published online 2 February 2026)

**Corresponding authors** S. Ono and N. Fujii, Institute of Health and Sport Sciences, University of Tsukuba, Tsukuba City, Ibaraki 305-8574, Japan. Email: ono.seiji.fp@u.tsukuba.ac.jp and fujii.naoto.gb@u.tsukuba.ac.jp

**Key points**

- Eye movements, such as fixations and saccades, are essential for visual stability and object tracking in daily life. Hyperventilation, which causes hypocapnia and cerebral hypoperfusion, can occur during physiological or psychological stress or in individuals with chronic disease.
- In this study we demonstrated that acute voluntary hypocapnic, but not normocapnic, hyperventilation impaired visual fixation variables, including the number of fixations and saccades, fixation duration and scanpath length.
- Both hypocapnic and normocapnic hyperventilation impaired the latency of anti-saccades, with hypocapnic hyperventilation causing a more pronounced impairment.
- We conclude that (1) hypocapnia induced by hyperventilation may impair oculomotor responses by weakening visual fixation and saccadic control, and (2) hyperventilation itself can also impair saccadic control.
- These oculomotor impairments associated with hypocapnic hyperventilation might increase the risk of injury or death in tasks that require precise visuomotor coordination.

## Introduction

Eye movements play a crucial role in capturing visual objects on the fovea of the retina, contributing to the acquisition of a clear and stable view. Visual fixation and saccades function to minimise ocular drift and rapidly redirect the line of sight to align objects of interest with the fovea, respectively. These functions are closely associated with a broad range of cognitive processes, including memory (Brown et al., 2004), attention (Zhao

et al., 2012), prediction (Shelhamer & Joiner, 2003) and decision-making (Seideman et al., 2018). Consequently, impaired eye movement function may increase the risk of injury and mortality in activities requiring precise visuomotor coordination, such as driving, operating machinery, working at heights and performing medical procedures, including surgery. These eye movements are also coordinated with hand and body movements to support goal-directed behaviour (de Brouwer et al., 2021), even in highly practiced tasks such as making tea (Land et al., 1999). Therefore, eye movements are precisely controlled by the central nervous system to acquire clear and stable visual information, and eye movement measurements are also used as neurophysiological biomarkers. However, physiological factors that influence eye movement control remain to be fully elucidated.

Breathing is tightly regulated to meet metabolic demands. However, environmental stressors such as heat exposure (Fujii et al., 2008; Gaudio Jr & Abramson, 1968; Katagiri et al., 2024) and high altitude (Lenfant & Sullivan, 1971), as well as psychological stressors like anxiety (Missri & Alexander, 1978; Pfeffer, 1978), can elicit hyperventilation – an increase in ventilation that exceeds metabolic requirements. In addition, certain chronic disease conditions, including cirrhosis (Karetzky & Mithoefer, 1967) and encephalitis (Harrop & Loeb, 1923), have also been reported to trigger hyperventilation. Hyperventilation may provoke abnormal eye movements in some individuals with specific inborn errors of metabolism (Koens et al., 2022). Furthermore, hyperventilation leads to a reduction in arterial partial pressure of carbon dioxide (i.e., hypocapnia), which in turn decreases cerebral blood flow (Ito et al., 2003). Compared with healthy controls, individuals with Alzheimer's disease, Huntington's disease and schizophrenia who exhibit reduced cerebral blood flow in specific brain regions (Goozée et al., 2014; Kisler et al., 2017; Montoya et al., 2006) also demonstrate abnormal oculomotor function (Anderson & MacAskill, 2013; Morita et al., 2020; Opwonya et al., 2022). Therefore, hyperventilation-induced hypocapnia may impair fixation and saccadic function, although this hypothesis requires direct investigation.

The purpose of this study was to investigate the effect of hyperventilation-induced hypocapnia on fixation and saccades in resting healthy young individuals. To address this, we employed two complementary oculomotor paradigms: a free-viewing task and an anti-saccade task, both of which are widely used in clinical neuroscience research (Anderson & MacAskill, 2013; Morita et al., 2020; Opwonya et al., 2022). The free-viewing task reflects natural fixation and gaze shift patterns that underlie visual exploration and scene perception. It engages large-scale visual and attentional networks distributed reflecting higher-order brain functions, thereby serving as an ecologically valid probe of visuomotor coordination (Mirpour & Bisley, 2021; Xiao et al., 2024; Zhang et al., 2024; Zhang et al., 2025). Furthermore, the anti-saccade task requires the suppression of a reflexive eye movement toward a visual stimulus, the initiation of a voluntary saccade in the opposite direction, and short-term memory for spatial location. Thus, it involves top-down inhibitory and executive control mechanisms mediated by the dorsolateral prefrontal cortex, frontal eye fields, supplementary eye fields, basal ganglia and superior colliculus (Coe & Munoz, 2017; Gaymard et al., 1999; Pierrot-Deseilligny et al., 2003; Schlag-Rey et al., 1997).

## Methods

### Ethical approval

This study was approved by the Human Subjects Committee of the University of Tsukuba (no. 023–114), in accordance with the latest version of *Declaration of Helsinki* except database registration. Written informed consent was obtained from all volunteers prior to their participation in this study.

### Participants

Thirteen participants (eight women) volunteered for the present study. They were (means $\pm$ SD) 23.1 $\pm$ 2.4 years of age, 1.64 $\pm$ 0.09 m in height and 60.4 $\pm$ 16.5 kg in body mass. All participants were healthy non-smokers with normal or corrected-to-normal vision and no history of neurological and oculomotor disorders. They were also not taking any prescription medication.

The sample size for this experiment was determined using *a priori* power analysis conducted with G*Power software (Version 3.1.9.7). The analysis was based on a within-subjects analysis of variance (ANOVA) with six measurements using $\alpha$ level of 0.05, a power of 0.80, and an effect size (Cohen's $f$) of 0.40. A correlation coefficient was set at 0.50 for repeated measures, and a non-sphericity correction $\varepsilon$ was set at 1.00. The effect size was derived from a previous study that examined the influence of manipulated visual stimuli on the conscious awareness of emotion using a within-subjects design (Sato & Yoshikawa, 2023), which was considered sufficiently similar in experimental structure to inform our estimate. Although the present study employed generalised linear mixed models (GLMM) for statistical analysis to account for subject-level variability, the ANOVA-based estimation was considered a practical approximation in the absence of prior GLMM-specific data. The calculated minimum sample size was eight, and we recruited more than eight participants to allow for potential attrition.

The menstrual cycle was not accounted for in female participants, as previous studies reported mixed results regarding the influence of sex or sex hormones on eye movement responses (Giddey et al., 2020; Mack et al., 2020; Wolohan et al., 2013).

### Familiarisation session

Prior to the experimental session, participants practiced the eye movement task and voluntary hyperventilation procedure (see below for details) in an environmental chamber (Fuji Medical Science Co., Ltd, Chiba, Japan) maintained at 25°C and 50% relative humidity. Participants were instructed to perform voluntary hyperventilation by increasing respiratory frequency to 30 breaths/min, guided by a metronome, whilst maintaining an individually determined tidal volume to reduce end-tidal carbon dioxide partial pressure to 20 mmHg.

### Experimental session

The experimental session was conducted within 1 week after the completion of the familiarisation session. Participants were instructed to abstain from consuming caffeine and alcohol for at least 24 h and from eating any food for at least 2 h prior to the experimental session. On the day of the experimental session, body mass was measured by weighing scales and moved to an environmental chamber regulated at 25°C and 50% relative humidity wherein they were seated. Subsequently, a firm neck pillow for fixing head position and a mask for measuring respiratory gases were attached. Then a transcranial Doppler probe for measuring middle cerebral artery mean blood velocity, cuffs for measuring arterial blood pressure and a probe for measuring percutaneous oxygen saturation, an index of arterial oxygen saturation, were attached to the left side of their temporal window, right upper arm and their left middle finger, and forehead, respectively. A chest strap was also attached for monitoring heart rate. After instrumentation, the experimental procedure, as illustrated in Fig. 1, was commenced under three different breathing conditions (see 'Breathing intervention' section). Initially, in all conditions, participants rested for 5 min, followed by the eye movement task lasting approximately 5–8 min whilst maintaining spontaneous breathing. Afterward, they breathed in a designated way for 15 min, and performed the eye movement task again whilst maintaining the designated breathing method. Measurements under the three breathing conditions were conducted in random order, with an interval of at least 30 min between each condition. Throughout the eye movement task, the experimental room was dimly lit to ensure the accuracy of the eye movement measurements.

### Breathing intervention

Three breathing interventions were (1) spontaneous breathing (control), (2) hypocapnic hyperventilation, and (3) normocapnic hyperventilation. In both conditions (2) and (3), participants were required to breathe at a rate of 30 breaths/min guided by a metronome whilst maintaining an individually determined tidal volume based on visual feedback displayed in a computer screen. As participants needed to focus on a separate computer screen during the eye movement task, visual feedback of tidal volume was impossible; instead an experimenter provided verbal feedback on tidal volume levels as needed. Participants breathed through a mask equipped with corrugated tubes attached to a three-way valve for measuring respiratory gases regardless of breathing conditions. In conditions (2) and (3), nitrogen gas was constantly added to inspired air to minimise an increase in end-tidal oxygen pressure associated with voluntary hyperventilation. In condition (3), carbon dioxide was also added to inspired air in order to maintain end-tidal carbon dioxide partial pressure at levels comparable to normocapnia. The concentrations of nitrogen and carbon dioxide in inspired air were manually adjusted using a gas flowmeter, with breath-by-breath data monitored via a metabolic cart (AE310s, Minato Medical Science, Osaka, Japan). Inspiratory oxygen and carbon dioxide concentrations during hypocapnic and normocapnic hyperventilation were $16.8 \pm 0.4\%$ oxygen and $0.4 \pm 0.1\%$ carbon dioxide, and $16.5 \pm 0.3\%$ oxygen and $3.9 \pm 0.4\%$ carbon dioxide, respectively. These oxygen concentrations were intentionally adjusted to maintain normoxic conditions, since hyperventilation can slightly elevate arterial oxygen partial pressure.

### Eye movement task

Visual stimuli were generated using the Psychophysics Toolbox in MATLAB R2024a (MathWorks, Natick, MA, USA) and presented on a 21.5-inch IPS LCD monitor (VP229HV, ASUSTeK Computer Inc., Taipei, Taiwan) at a viewing distance of 57 cm. The monitor had a refresh rate of 75 Hz, a spatial resolution of $1920 \times 1080$ pixels, and a display response time of $\sim 1$ ms. In the present study, free-viewing and anti-saccade tasks were conducted (Fig. 1). During the free-viewing task, participants were instructed to freely observe a stationary landscape image for 5 s. Each trial was preceded by a three-count displayed on the monitor, and the task was repeated 10 times with a different image presented in each trial, resulting in a total viewing time of 50 s. During the anti-saccade task, a green

circle with a diameter of 5 mm was presented as a visual target at the centre of the monitor (at a viewing angle of 0°) for 1–4 s. Subsequently, the green circle was shifted to a viewing angle of 10°, either to the left or to the right side of the monitor, and was presented for 1 s. Participants were asked to keep their gaze on the target whilst it was presented at the centre of the monitor. They were also required to move their gaze to the mirror position of the green circle as quickly as possible, without moving their gaze towards the green circle, when it shifted. Specifically, participants had to move their gaze to a viewing angle of 10° to the left side of the monitor when the green circle shifted to a viewing angle of 10° to the right side of the monitor. In total, 40 anti-saccade trials lasting ∼200 s were completed, with the green circle shifting to a viewing angle of 10° on the left side of the monitor in half of the trials.

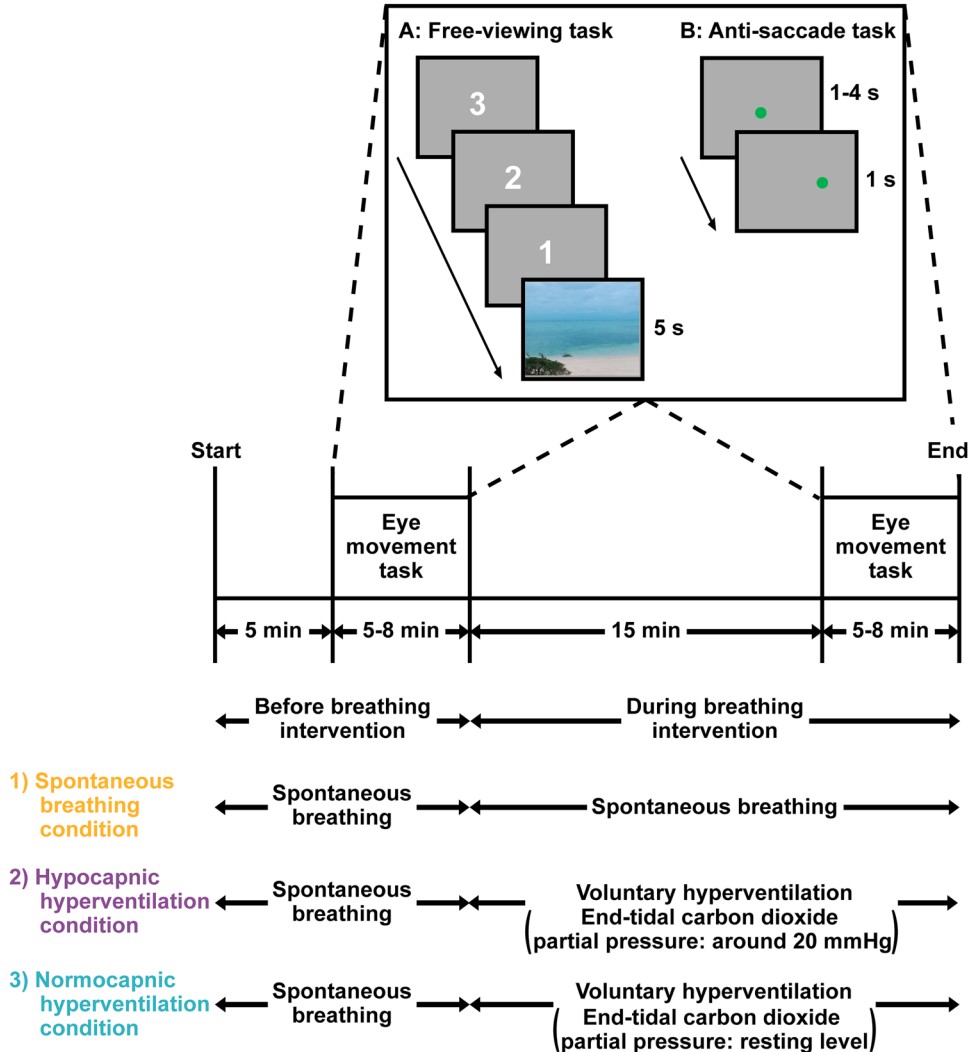

**Figure 1. Experimental protocol**
The eye movement task (free-viewing and anti-saccade tasks) was conducted during both spontaneous breathing and each breathing intervention. The breathing interventions were: (1) spontaneous breathing (control) condition, (2) voluntary hypocapnic hyperventilation, characterised by a breathing rate of 30 breaths/min and a reduction in end-tidal carbon dioxide partial pressure to approximately 20 mmHg, and (3) voluntary normocapnic hyper-ventilation condition, conducted at a breathing rate of 30 breaths/min, with end-tidal carbon dioxide partial pressure maintained at normocapnic levels. During free-viewing task (*A*), a stationary landscape image was presented for 5 s, followed by three counts on the monitor. Thereafter, participants were asked to view the image freely. During the anti-saccade task (*B*), a green circle as a visual target was presented as the centre of the monitor for 1–4 s, then shifted to a viewing angle of 10° on the left or right side of the monitor. Participants were required to move their gaze to the mirror position of the shifted target as quickly as possible.

### Additional experimental session

To verify whether participants develop respiratory alkalosis during hypocapnic hyperventilation, we conducted an additional experimental session in the environmental chamber maintained at 25°C and 50% relative humidity. Six participants (three men and three women) who joined the experimental session took part in the additional experimental session. Their age was 22.5 ± 1.4 years, with a height of 1.68 ± 0.08 m and a body mass of 60.4 ± 16.5 kg (means ± SD). All participants were told to adhere to the same contraindications provided during the main experiment. Upon arrival, a mask for measuring respiratory gases was attached whilst the participants remained seated. Participants then underwent the same experimental procedure of hypocapnic hyperventilation and normocapnic hyperventilation conditions without the eye movement task. Both trials were conducted in random order on the same day, with an interval of at least 30 min between the two.

### Measurements and data analysis

**Eye movement variables.** Eye movements of a single eye (either left or right) were recorded using a video-based eye-tracking system (Matsuda et al., 2017). As the eye movements measured in this study can be classified as conjugate behaviour (Leigh & Zee, 2015), no substantial differences in eye movements were expected between the left and right eyes.

The eye-tracking system, operated with iRecHS2 software (Version 0.663; National Institute of Advanced Industrial Science and Technology, Ibaraki, Japan; system latency: ∼6 ms at 250 Hz sampling rate), utilised a video camera (Grasshopper 3, S3-U3-41C6NIR, Teledyne FLIR LLC, OR, USA) set to a sampling rate of 250 Hz to capture the infrared light reflection from the cornea and the dark image of the pupil. Eye position signals, which can be measured with an accuracy of <0.2° (Matsuda et al., 2017), were detected by identifying the centres of the corneal reflection and the pupil. These signals were digitised at a sampling rate of 250 Hz with 16-bit precision using a data acquisition unit (Micro1401-2, Cambridge Electronic Design, Cambridge, UK).

Prior to the eye movement task, the system was calibrated by requiring participants to fixate sequentially on five white circles (diameter: 3 mm) presented on a grey background of a monitor (VP229HV) at visual angles of 0°, ±10° horizontally and ±10° vertically. Calibration was validated by confirming that eye positions during fixation on these targets remained within ±1° of the expected visual angle, as assessed visually by the experimenter.

Eye-movement data were analysed using MATLAB R2024a (MathWorks). Eye position data were filtered with a second-order Butterworth low-pass filter with a 30-Hz passband. Eye velocity and acceleration were calculated by the first and second derivatives of the position arrays, and were filtered with a second-order Butterworth low-pass filter with a 30-Hz passband. Prior to analysing each trial of the eye movement task, data containing blinks, closed eyes and unphysiological eye movements (e.g., eye velocity exceeding 1000°/s or eye acceleration exceeding 100,000°/s$^2$) were removed as recommended (Nyström & Holmqvist, 2010).

In the free-viewing task, saccades and fixations were detected using an eye velocity-based algorithm (Nyström & Holmqvist, 2010). Specifically, time points at which eye velocity exceeded a threshold of 100°/s, as suggested by Nyström and Holmqvist (2010), were first identified as candidate saccades. For each detected saccade, we first identified the time point at which the velocity trace exceeded a predefined threshold. The onset of the saccade was then defined by tracing the velocity signal backward from this threshold crossing until reaching the preceding local minimum. Likewise, the offset was defined by tracing the velocity signal forward from the threshold crossing until the subsequent local minimum was reached. In other words, the onset corresponds to the last minimum before the velocity rise, and the offset corresponds to the first minimum after the velocity decrease. Further details regarding this parameterisation procedure can be found in the original paper (Nyström & Holmqvist, 2010).

Only events with a duration of at least 12 ms were classified as saccades. Fixations were defined as periods between saccades with a minimum duration of 40 ms. Subsequently, the number of fixations and saccades was computed during each 5-s visual search trial. Saccade duration was defined as the time interval from saccade onset to offset, and saccade amplitude was quantified as the distance between eye positions at onset and offset. Finally, scanpath length was calculated as the cumulative sum of saccade amplitudes within one trial. Across participants, data loss during the trials was 6.3 ± 5.3%, primarily due to brief tracking interruptions such as blinks or momentary loss of pupil detection.

In the anti-saccade task, pro-saccades (error responses defined as pro-saccade errors) and anti-saccades (correct responses) were identified using the same algorithm employed in the free-viewing task. The presence or absence of error responses was binarized and classified. Additionally, the latency of pro-saccade errors and anti-saccades was calculated, defined as the time from the visual target's shift to a viewing angle of 10° to the onset of the respective response. The peak velocity and gain of anti-saccades and pro-saccade errors were computed, with gain defined as the ratio of anti-saccade or pro-saccade amplitude to the target amplitude (10°). Trials were excluded if response latencies were <80 ms, if pro-saccade error latencies exceeded 300 ms or if anti-saccade latencies

exceeded 800 ms. Across participants, the proportion of excluded (invalid) trials was $13.7 \pm 7.3\%$.

**Physiological variables.** Minute ventilation, tidal volume, respiratory frequency, end-tidal carbon dioxide partial pressure, end-tidal oxygen partial pressure, oxygen uptake and carbon dioxide output were measured breath-to-breath using a metabolic cart (AE310s, Minato Medical Science, Osaka, Japan). Prior to the measurement, the flow sensor was calibrated using an appurtenant calibration syringe with a known volume of 2 litres and reference gases at known concentrations (oxygen: 15.10% and carbon dioxide: 5.01%). Respiratory variables measured were averaged every 10 s.

Middle cerebral artery mean blood velocity was measured using a transcranial Doppler ultrasound device (EZ-Dop, Compumedics Germany GmbH, Singen, Germany). A 2-MHz Doppler probe was affixed with an adjustable headband over the temporal window on the left side, and the signal was collected at a depth of 46–52 mm. The signal was recorded at a sampling rate of 200 Hz using a data acquisition system (PowerLab 8/35, ADInstruments, Dunedin, New Zealand). Due to technical difficulties, the blood flow velocity data could not be successfully collected or analysed for one participant under condition (2) and for two participants under condition (3).

Beat-to-beat arterial blood pressure was estimated from the left middle finger, positioned on a table fixed at heart level using finger photoplethysmography (Finometer, Finapres Medical Systems, Enschede, the Netherlands). Arterial blood pressure was also measured using an automated sphygmomanometer (TM-2580, A&D Company, Ltd, Tokyo, Japan), with data obtained during the 5-min rest with spontaneous breathing used for correcting beat-to-beat finger arterial blood pressures. Mean arterial pressure was calculated as diastolic pressure plus one-third of the pulse pressure. Heart rate was recorded every second using a heart rate monitor (V800, Polar Electro, Kempele, Finland). Stroke volume was estimated using the Modelflow method with assistance from the finger photoplethysmography (Finometer). Cardiac output was calculated as the product of stroke volume and heart rate. Due to technical difficulties, stroke volume, cardiac output and arterial blood pressure data could not be successfully collected for one participant during the breathing intervention under condition (2). Additionally, heart rate and stroke volume data could not be successfully collected or analysed for one participant during the 5-min rest under condition (3).

Percutaneous oxygen saturation was measured using forehead pulse oximetry (Nellcor N-595, Medtronic, Minneapolis, MN, USA) and was recorded at a sampling rate of 200 Hz using a data acquisition system (PowerLab 8/35, ADInstruments). Due to technical difficulties, percutaneous oxygen saturation data could not be successfully collected or analysed for one participant during the breathing intervention under condition (2).

In an additional experiment only, blood samples were collected from the fingertip at two time points: immediately prior to voluntary hyperventilation and 15 min into voluntary hyperventilation with each measurement conducted twice. All collected blood samples were entered into a cartridge (i-STAT G3+, Abbott Laboratories, Abbott Park, IL, USA) of a portable blood gas analyser (i-STAT 1 analyser, Abbott Laboratories). Subsequently, pH, partial pressure of carbon dioxide and oxygen, bicarbonate ions, total carbon dioxide and oxygen saturation in the blood were measured. Data averaged across two blood samples were used for data analysis.

**Perceptual variable.** A subjective feeling of eye fatigue was quantified using a visual analogue scale (VAS), where participants indicated their level of fatigue on a 10 cm line ranging from 0 (no fatigue) to 100 (worst experienced fatigue). A subjective effort of breathing was assessed using an 11-point scale, ranging from 0 (no effort at all) to 10 (maximum effort). Both perceptions were measured immediately prior to the execution of the eye movement task.

### Statistical analysis

GLMM or linear mixed models (LMM), implemented in R (Version 4.4.2), were employed to analyse the eye movement variables, accounting for the order of the three breathing conditions and interindividual variability. To minimise the influence of outliers, we excluded the top and bottom 1% of values in each dataset based on their empirical distribution during preprocessing. For GLMMs, the appropriate distribution family and link function were selected based on the response variable type (eye movement variables). For LMMs, normality of residuals was assessed. In GLMMs and LMMs, these modelling decisions were determined by visual inspection of histograms and Q–Q plots, and the Shapiro–Wilk test. GLMMs and LMMs were fitted using the *lme4* package. The significance of fixed effects was evaluated with Type III Wald $\chi^2$ tests using the *car* package. We specified GLMMs or LMMs as follows:

$$Y_{ij} = \beta_0 + \text{intervention}_{ij} + \text{phase}_{ij} + \text{order}_{ij}$$
$$+ \left(\text{intervention} \times \text{phase}\right)_{ij} + b_{0i}$$

where $Y_{ij}$ denotes the response variables for participant $i$ at observation $j$. The fixed predictors included intervention, phase, and order, treated as categorical factors.

The intervention had three levels (spontaneous breathing, hypocapnic hyperventilation and normocapnic hyperventilation) and phase had two levels (before and during breathing intervention) as fixed predictors. The fixed predictors also included order (three levels: first, second and third), representing the order in which the three breathing conditions were administered to account for potential carry-over effects. Moreover, the interaction between the intervention and phase was incorporated into the models. $\beta_0$ denotes the fixed intercept, and the participants were included in the models as a random intercept ($b_{0i}$) to account for interindividual variability.

Estimated marginal means (EMMs) were computed for each combination using the *emmeans* package. When the main effect of intervention or the interaction between intervention and phase was significant, these EMMs were then used for multiple comparison tests (a *z*-test for GLMM and Student's *t*-test for LMM), with *P*-values adjusted by the Benjamini–Hochberg procedure.

In order to analyse physiological and perceptive variables in the experimental session, we conducted a repeated-measures two-way ANOVA with factors of intervention (three levels as noted above) and phase (two levels as noted above), using the *ez* package. In the additional experimental session, the same ANOVA was conducted for blood gas variables with factors of intervention (two levels: hypocapnic hyperventilation and normocapnic hyperventilation) and phase (two levels as noted above). When a significant main effect of the intervention or a significant interaction between the intervention and phase was found, *post hoc* pairwise comparisons were conducted using a *t*-test with *P*-values adjusted by the Benjamini–Hochberg procedure.

For all analyses, the level of statistical significance was set at $P < 0.05$. Physiological variables are presented as means with individual data. Eye-movement variables derived from the mixed-effects models are shown as estimated marginal means with individual data. All values reported in the tables are expressed as the mean ± SD.

## Results

### Minute ventilation, end-tidal carbon dioxide partial pressure, cerebral blood flow index and arterial oxygen saturation

As designed, minute ventilation increased during both hypocapnic and normocapnic hyperventilation relative to pre-hyperventilation levels (Fig. 2*A*). As expected, end-tidal carbon dioxide partial pressure decreased to approximately 20 mmHg during hypocapnic hyperventilation (Fig. 2*B*). End-tidal carbon dioxide partial pressure slightly increased during normocapnic hyperventilation, whilst under spontaneous breathing conditions subtly decreased relative to pre-hyperventilation level; however, these changes were physiologically negligible (+1.1 ± 1.5 and −0.3 ± 0.4 mmHg, respectively, Fig. 2*B*). Middle cerebral artery mean blood velocity decreased markedly under hypocapnia (∼23.7 ± 9.9 cm/s, Fig. 2*C*) and modestly during spontaneous breathing (∼1.7 ± 2.1 cm/s, Fig. 2*C*) and under normocapnic hyperventilation (∼8.5 ± 8.4 cm/s, Fig. 2*C*). Percutaneous oxygen saturation exceeded 95% in all phases (Fig. 2*D*).

### Oculomotor responses in the free-viewing task

Hypocapnic hyperventilation reduced the number of fixations and saccades, as well as scanpath length, whilst increasing the duration of fixation (Fig. 3*A–C* and *F*). In contrast, these changes were not mediated by normocapnic hyperventilation (Fig. 3*A–C* and *F*).

### Oculomotor responses in the anti-saccade task

Anti-saccade latency increased during both normocapnic (+10.9 ms, 95% CI: 2.6–19.2 ms) and hypocapnic (+22.3 ms, 95% CI: 13.0–31.5 ms) hyperventilation, with the effect being more pronounced during hypocapnic hyperventilation (Fig. 4*B*). Anti-saccade gain did not change under three interventions, although the interaction was significant (Fig. 4*D*). No interactions were found for error rate (Fig. 4*A*) and peak velocity of anti-saccades (Fig. 4*C*), as well as erroneous pro-saccade variables (Fig. 4*E–G*).

### Ventilatory responses

Tidal volume and respiratory frequency increased during both normocapnic and hypocapnic hyperventilation, with greater increases observed under normocapnia (Table 1). End-tidal oxygen partial pressure remained unchanged in both conditions. Oxygen uptake increased during hypocapnic hyperventilation, whilst carbon dioxide output increased under both conditions (Table 1).

### Cardiovascular responses and perceptions

No interactions were found for heart rate, stroke volume or cardiac output (Table 2). There was a main effect of intervention on heart rate, but post hoc comparisons indicated no differences between any of the interventions (Table 2). Although a significant interaction was detected for mean arterial pressure, *post hoc* comparisons revealed no difference between interventions or between before and during intervention phases (Table 2).

Subjective feelings of eye fatigue and breathing effort increased during both normocapnic and hypocapnic hyperventilation (Table 2).

### Blood gas variables

pH increased during hypocapnic, but not normocapnic, hyperventilation (Table 3). Partial pressure of carbon dioxide and oxygen, and bicarbonate ions decreased during hypocapnic hyperventilation, but these changes were not observed during normocapnic hyperventilation (Table 3). Neither form of hyperventilation produced

changes in total carbon dioxide and oxygen saturation (Table 3).

## Discussion

We demonstrated that voluntary hypocapnic hyperventilation, which was accompanied by a marked reduction in middle cerebral artery mean blood velocity, reduced the numbers of fixations and saccades, increased fixation duration, and shortened the scanpath length during the free-viewing task. By contrast, these alterations were not elicited by voluntary normocapnic hyper-

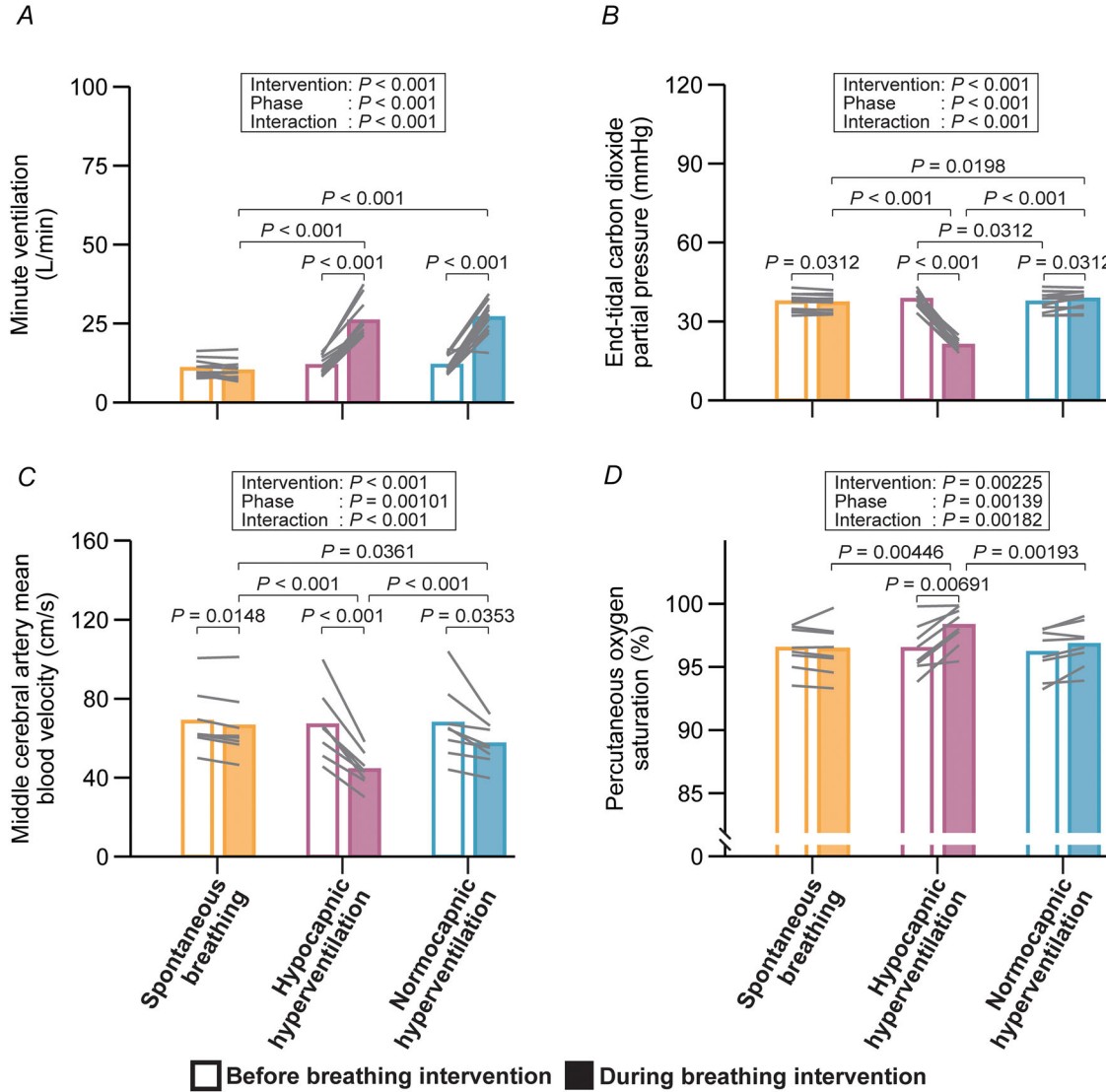

**Figure 2. Respiratory, cerebral blood flow and oxygen saturation responses during the eye movement task**
Minute ventilation (*A*), end-tidal carbon dioxide partial pressure (*B*), middle cerebral artery mean blood velocity (*C*), and percutaneous oxygen saturation (*D*) measured before and during breathing intervention (see Fig. 1 for details). Bar graphs represent mean values for each combination. Grey lines indicate individual data. A repeated two-way ANOVA was performed, followed by *post hoc* pairwise comparisons using *t*-tests with *P*-values adjusted by the Benjamini–Hochberg procedure.

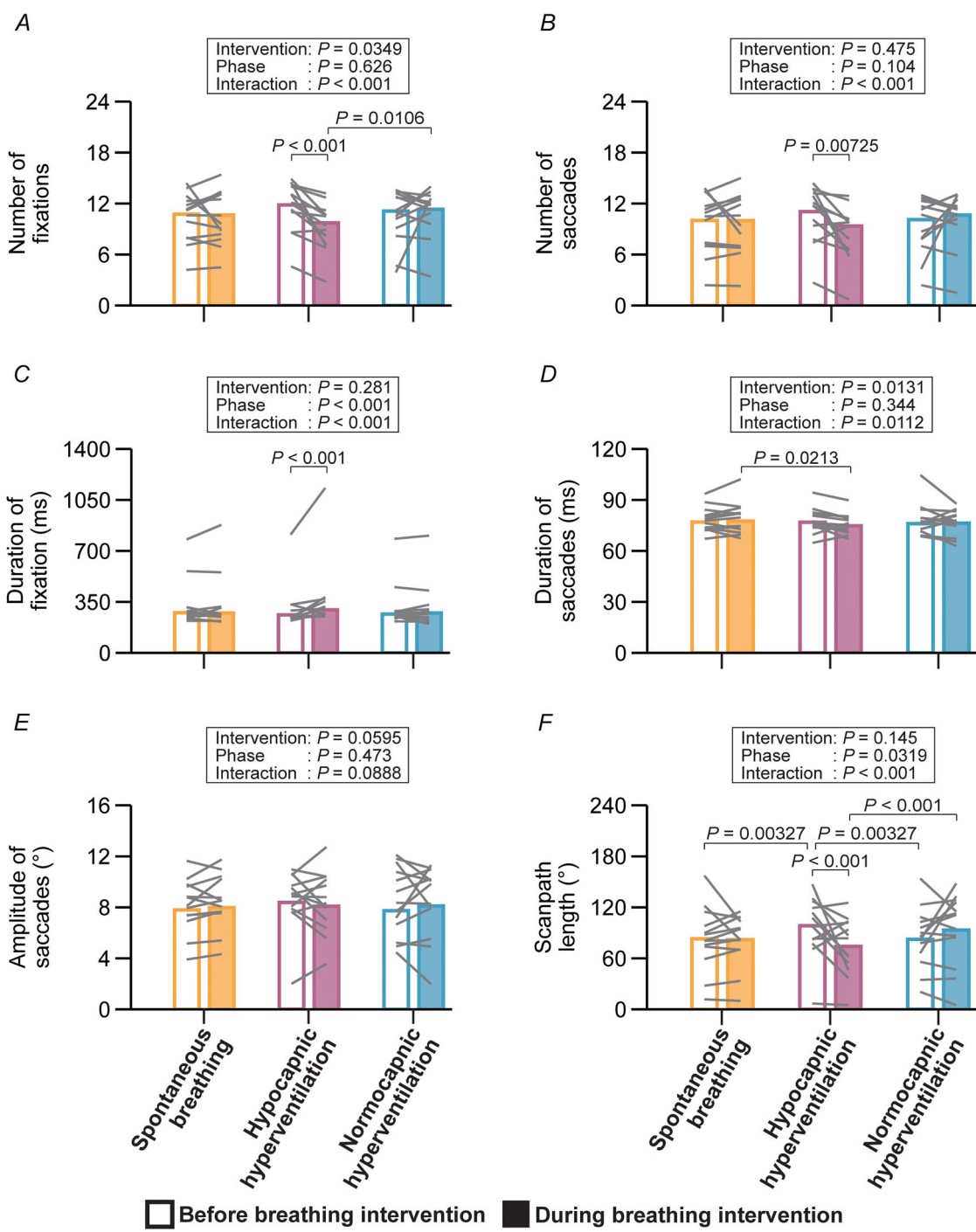

**Figure 3. Oculomotor responses for the free-viewing task**
Number of fixations (*A*), number of saccades (*B*), duration of fixation (*C*), duration of saccades (*D*), amplitude of saccades (*E*), and scanpath length (*F*) measured before and during breathing intervention. The number of fixations and saccades was modelled using a Poisson distribution with a log link function; fixation and saccade durations and amplitudes were modelled using a gamma distribution with a log link function; scanpath length was modelled with a normal distribution. Bar graphs represent estimated marginal means (EMMs) for each combination and grey lines indicate individual data. Fixed effects were evaluated with Type III Wald $\chi^2$ tests. EMMs were employed as *post hoc* tests for multiple comparisons (*z*-tests for GLMM and *t*-tests for LMM), with *P*-values adjusted by the Benjamini–Hochberg procedure.

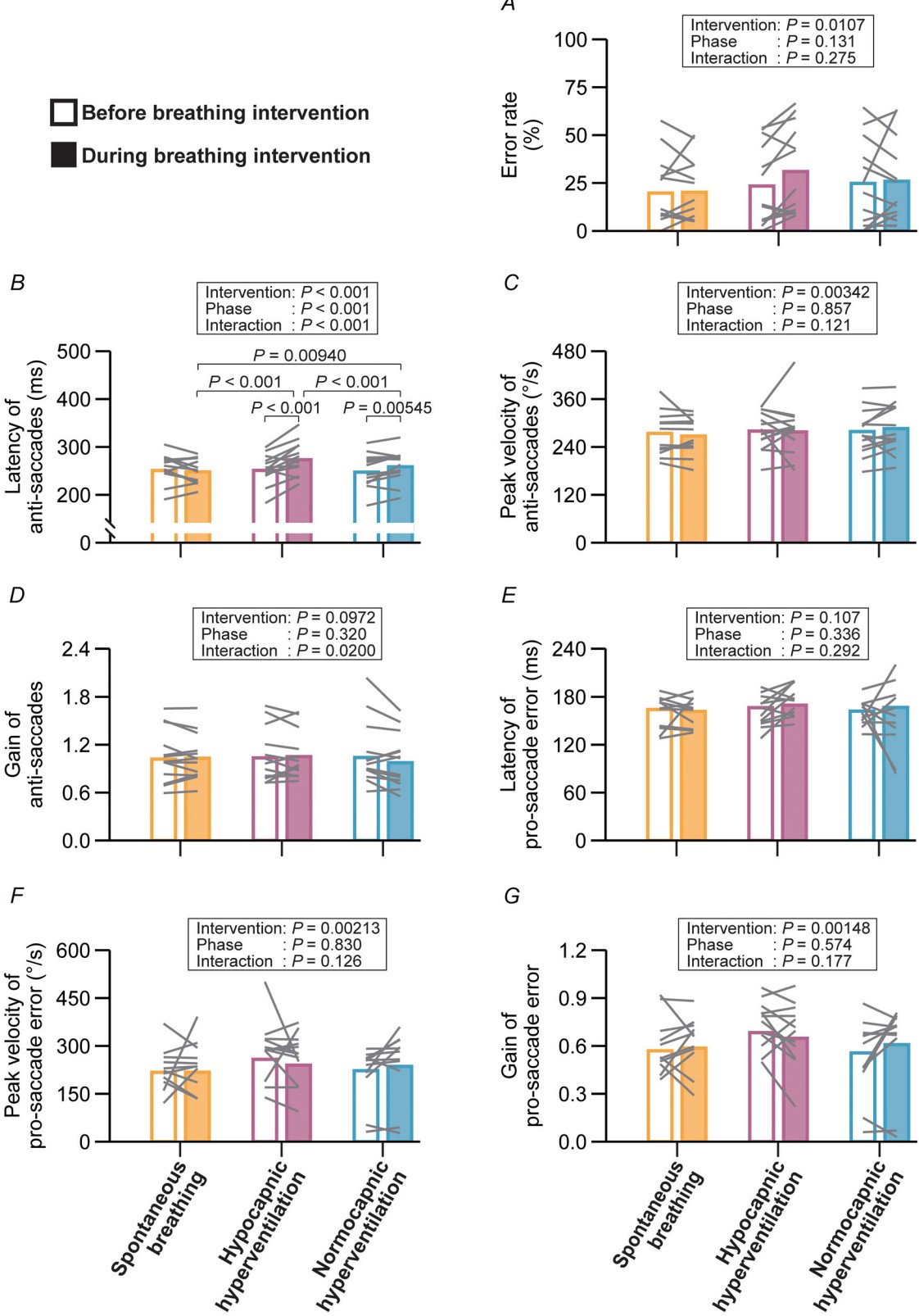

**Figure 4. Oculomotor responses for the anti-saccade task**
Error rate (*A*), latency (*B*), peak velocity (*C*) and gain (*D*) of anti-saccades, as well as latency (*E*), peak velocity (*F*) and gain (*G*) of pro-saccade errors measured before and during breathing intervention. Error rate, the peak

velocity and gain of pro-saccade errors were modelled using a normal distribution. Latency, peak velocity and gain of anti-saccades, as well as the latency of pro-saccade errors, were modelled using a Gamma distribution with a log link function. Bar graphs represent EMMs for each combination, and grey lines indicate individual data. Fixed effects were evaluated with Type III Wald $\chi^2$ tests. EMMs were employed as *post hoc* tests for multiple comparisons (*z*-tests for GLMM and *t*-tests for LMM), with *P*-values adjusted by the Benjamini–Hochberg procedure.

ventilation. Additionally, both voluntary normocapnic and hypocapnic hyperventilation prolonged anti-saccade latency during the anti-saccade task, with hypocapnic hyperventilation exhibiting greater changes. We show that hyperventilation induced hypocapnia impairs oculomotor responses by modulating fixation and saccadic control in healthy young adults. In addition, hyperventilation itself impairs saccadic control independently of hypocapnia.

### Effect of hypocapnic hyperventilation on oculomotor responses

In the free-viewing task, hypocapnic hyperventilation led to fewer fixations and saccades, shorter scanpath length, and longer fixation durations (Fig. 3). By contrast,

these changes were not mediated by normocapnic hyperventilation (Fig. 3). These results suggest that the hypocapnia induced by voluntary hyperventilation impairs visual search behaviour in healthy young adults. Visual search behaviour is known to involve visual attention based on working memory, such that higher memory load is associated with prolonged fixation durations (Meghanathan et al., 2015; Peterson et al., 2008) and fewer fixations, and shorter scanpath length (Cronin et al., 2020). Reduced middle cerebral artery mean blood velocity resulting from hypocapnic hyperventilation (Fig. 2C) may reflect diminished blood and oxygen delivery to the prefrontal and parietal cortices (Fig. 5), which are critically involved in working memory (Sarnthein et al., 1998; Todd & Marois, 2005). Consequently, neural activity within these areas may be

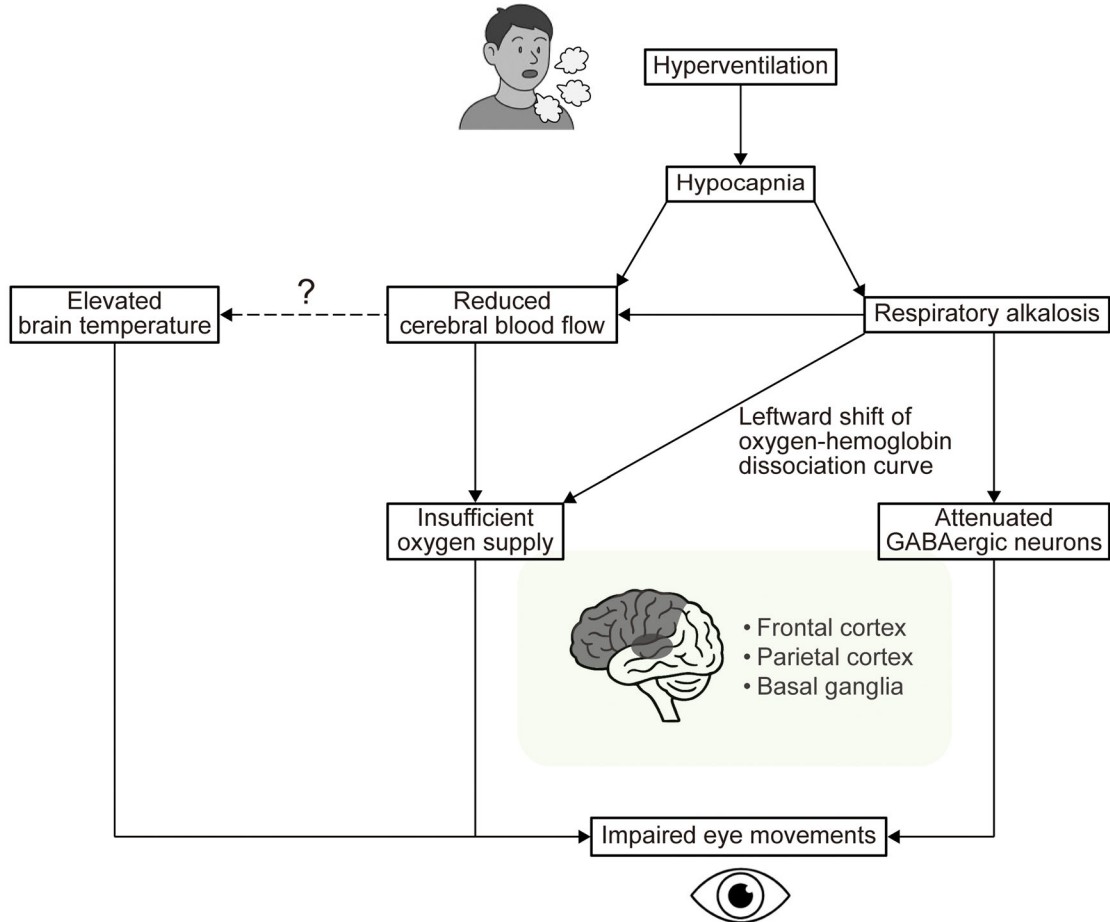

**Figure 5. Overview of the effects of hypocapnia on oculomotor function**

**Table 1. Ventilatory responses before and during the eye movement task**

| | Spontaneous breathing | | Hypocapnic hyperventilation | | Normocapnic hyperventilation | | ANOVA *P*-value | | |
| --- | --- | --- | --- | --- | --- | --- | --- | --- | --- |
| | Before | During | Before | During | Before | During | Intervention | Phase | Interaction |
| Tidal volume (mL) | 561 ± 155 | 516 ± 171* | 613 ± 142† | 859 ± 169***, †††, ## | 599 ± 165 | 946 ± 147***, ††† | *<0.001* | *<0.001* | *<0.001* |
| Respiratory frequency (breaths/min) | 19.4 ± 2.2 | 19.6 ± 2.8 | 19.3 ± 2.5 | 29.9 ± 0.4***, ††† | 19.8 ± 2.7 | 30.1 ± 0.8***, ††† | *<0.001* | *<0.001* | *<0.001* |
| End-tidal oxygen partial pressure (mmHg) | 107 ± 4 | 108 ± 4 | 106 ± 3 | 106 ± 5 | 107 ± 4 | 108 ± 4 | 0.932 | 0.197 | 0.270 |
| Oxygen uptake (mL/min) | 215 ± 53 | 193 ± 53* | 224 ± 59 | 245 ± 68†† | 220 ± 49 | 233 ± 49 | *0.0277* | 0.343 | *<0.001* |
| Carbon dioxide output (mL/min) | 175 ± 48 | 154 ± 48* | 185 ± 56 | 281 ± 87***, †††, ### | 176 ± 38 | 199 ± 43*, † | *<0.001* | *<0.001* | *<0.001* |

Values are means ± SD; *n* = 13. Before and During denote before breathing intervention and during breathing intervention, respectively. A repeated two-way ANOVA was performed, and *post hoc* pairwise comparisons were conducted using *t*-tests with *P*-values adjusted by the Benjamini–Hochberg procedure. Italic denotes statistically significant values. *$P$ < 0.05 *versus* before breathing intervention; ***$P$ < 0.001 *versus* before breathing intervention; †$P$ < 0.05 *versus* spontaneous breathing condition; ††$P$ < 0.01 *versus* spontaneous breathing condition; †††$P$ < 0.001 *versus* spontaneous breathing condition; ##$P$ < 0.01 *versus* normocapnic hyperventilation condition; ###$P$ < 0.001 *versus* normocapnic hyperventilation condition.

**Table 2. Cardiovascular variables and perceptions before and during the eye movement task**

| | Spontaneous breathing | | Hypocapnic hyperventilation | | Normocapnic hyperventilation | | ANOVA *P*-value | | |
| --- | --- | --- | --- | --- | --- | --- | --- | --- | --- |
| | Before | During | Before | During | Before | During | Intervention | Phase | Interaction |
| Heart rate (beats/min) | 61.9 ± 10.8 | 62.5 ± 10.5 | 62.8 ± 9.6 | 65.0 ± 12.9 | 63.3 ± 10.0 (*n* = 12) | 68.0 ± 15.1 | *0.0207* | 0.230 | 0.360 |
| Stroke volume (mL) | 85.1 ± 20.0 | 85.5 ± 20.7 | 83.5 ± 17.2 | 82.9 ± 16.1 (*n* = 12) | 79.9 ± 13.1 (*n* = 12) | 82.3 ± 16.7 | 0.577 | 0.211 | 0.643 |
| Cardiac output (L/min) | 5.2 ± 1.3 | 5.3 ± 1.5 | 5.2 ± 1.3 | 5.3 ± 1.4 (*n* = 12) | 5.0 ± 0.9 | 5.5 ± 1.1 | 0.162 | 0.0721 | 0.277 |
| Mean arterial pressure (mmHg) | 79.4 ± 10.4 | 81.1 ± 11.0 | 79.7 ± 10.6 | 74.2 ± 10.0 (*n* = 12) | 79.9 ± 9.5 | 82.8 ± 10.4 | 0.427 | 0.934 | *0.00798* |
| Feeling of eye fatigue | 11.0 ± 16.2 | 12.0 ± 17.8 | 8.7 ± 12.4 | 17.7 ± 17.6* | 9.8 ± 13.8 | 16.5 ± 20.1* | 0.884 | *0.0202* | *0.00786* |
| Effort of breathing | 0.5 ± 0.8 | 0.8 ± 0.9 | 0.5 ± 0.8 | 4.3 ± 1.1***, ††† | 0.5 ± 0.8 | 3.7 ± 1.4***, ††† | *<0.001* | *<0.001* | *<0.001* |

Values are means ± SD; *n* = 13 unless otherwise indicated. Before and during denote before breathing intervention and during breathing intervention, respectively. A repeated two-way ANOVA was performed, and *post hoc* pairwise comparisons were conducted using *t*-tests with *P*-values adjusted by the Benjamini–Hochberg procedure. Italic denotes statistically significant values. *$P$ < 0.05 *versus* before breathing intervention; ***$P$ < 0.001 *versus* before breathing intervention; †††$P$ < 0.001 *versus* spontaneous breathing condition.

**Table 3. Variables of blood gas analysis**

| | Hypocapnic hyperventilation | | Normocapnic hyperventilation | | ANOVA *P*-value | | |
| --- | --- | --- | --- | --- | --- | --- | --- |
| | Before | During | Before | During | Intervention | Phase | Interaction |
| pH | 7.37 ± 0.02 | 7.56 ± 0.02***, ### | 7.38 ± 0.02 | 7.40 ± 0.01 | *<0.001* | *<0.001* | *<0.001* |
| Partial pressure of carbon dioxide (mmHg) | 40.2 ± 4.5 | 20.9 ± 2.1***, ### | 39.4 ± 6.5 | 35.4 ± 2.9 | *<0.001* | *0.00177* | *<0.001* |
| Partial pressure of oxygen (mmHg) | 87.8 ± 13.0 | 74.0 ± 4.9*, # | 78.5 ± 4.8 | 85.8 ± 8.4 | 0.622 | 0.331 | *0.00944* |
| Bicarbonate ion (mmol/L) | 23.1 ± 2.0 | 18.9 ± 2.2**, # | 22.9 ± 2.7 | 21.9 ± 1.6 | *0.0151* | *0.00531* | *0.0441* |
| Total carbon dioxide (mmol/L) | 24.3 ± 2.1 | 19.5 ± 2.3 | 24.1 ± 3.2 | 23.0 ± 1.8 | *0.0129* | *0.00625* | 0.0594 |
| Oxygen saturation (%) | 95.8 ± 1.7 | 97.0 ± 0.8 | 95.0 ± 1.4 | 96.4 ± 1.2 | *0.0382* | 0.0668 | 0.713 |

Values are means ± SD; $n$ = 6. Before and during denote Before breathing intervention and during breathing intervention, respectively. A repeated two-way ANOVA was performed, and *post hoc* pairwise comparisons were conducted using *t*-tests with *P*-values adjusted by the Benjamini–Hochberg procedure. Italic denotes statistically significant values. *$P$ < 0.05 *versus* before breathing intervention; **$P$ < 0.01 *versus* before breathing intervention; ***$P$ < 0.001 *versus* before breathing intervention; #$P$ < 0.05 *versus* normocapnic hyperventilation condition; ###$P$ < 0.001 *versus* normocapnic hyperventilation condition.

attenuated, leading to impaired visual search behaviour. Elevated pH (>7.45) (Table 3) associated with hypocapnic hyperventilation shifts the oxygen–haemoglobin dissociation curve to the left, reducing oxygen unloading at peripheral sites, thereby further limiting oxygen delivery to tissues (Fig. 5). In a similar vein, compromised oculomotor control is commonly observed in neurological disorders such as Alzheimer's disease (Boz et al., 2023), Huntington's disease (Reyes-Lopez et al., 2024), and schizophrenia (Bansal et al., 2021). Notably, these disorders have also been associated with reduced cerebral perfusion (Goozée et al., 2014; Kisler et al., 2017; Montoya et al., 2006).

It is noteworthy to highlight that hypocapnic, but not normocapnic, hyperventilation reduced saccade frequency (Fig. 3*B*). Saccades are typically preceded by covert shifts of visual attention (Zhao et al., 2012), and these functions are tightly linked to activity in the lateral intraparietal area and frontal eye field (Bisley & Goldberg, 2010; Moore & Fallah, 2001). The lateral intraparietal area is thought to encode visual salience and guide attentional selection (Bisley & Goldberg, 2003; Colby et al., 1996; Gottlieb et al., 1998), whilst the frontal eye field is involved in saccade generation and attentional shifts (Moore & Fallah, 2001; Murthy et al., 2001; Thompson et al., 1996). Thus, hypocapnia associated with hyperventilation appears to decrease attentional deployment, contributing to reduced saccade generation based on visual attention. Furthermore, it is important to note that the shorter scanpath length was only due to hypocapnic hyperventilation (Fig. 3*F*). According to the biased competition theory, attentional selection relies on activity in the frontal and parietal cortices, which allocate resources across the visual field (Beck & Kastner, 2009; Desimone & Duncan, 1995). Hypocapnia linked with hyperventilation may attenuate activity in the frontal and parietal cortices (Fig. 5), narrowing attentional deployment and resulting in constrained visual exploration – a phenomenon resembling 'tunnel vision' (Mackworth, 1965).

In the anti-saccade task, hypocapnic hyperventilation prolonged anti-saccade latency relative to the normocapnic condition (Fig. 4*B*). This result suggests that hypocapnia mediated by hyperventilation impairs oculomotor performance under high cognitive load. The anti-saccade task engages the prefrontal cortex and basal ganglia (Leigh & Zee, 2015). In line with this, patients who have lesions in the frontal eye fields exhibit longer anti-saccade latencies (Gaymard et al., 1999). Moreover, disorders such as corticobasal degeneration and progressive supranuclear palsy, which affect the cerebrum and basal ganglia, are associated with impaired saccadic initiation (Rivaud-Pechoux et al., 2000). Reduced cerebral blood flow as indicated by middle cerebral artery mean blood velocity due to hypocapnia may attenuate the activity of the prefrontal

cortex and basal ganglia (Fig. 5), possibly underlying the impaired oculomotor performance. This notion is further supported by evidence showing that hypoxaemia, induced via hypoxic air inhalation, compromises executive and visuocognitive functions (Ochi et al., 2018). Moreover, alkalosis reportedly impairs the response to excitatory synaptic inputs and the ability to encode spikes of GABAergic neurons (Zhang et al., 2013), which are integral to inhibitory control and contribute to the suppression of pro-saccades in the anti-saccade task via basal ganglia circuits (Coe & Munoz, 2017). Thus, acute GABAergic malfunction resulting from respiratory alkalosis may also contribute to the deterioration of anti-saccade task performance (Fig. 5). Furthermore, reduced cerebral perfusion may elevate brain temperature (Nybo et al., 2002) (Fig. 5), which is associated with suppressed motor cortex activity and impaired executive functioning (Tan et al., 2024), possiblly attenuating anti-saccade task performance.

Interestingly, peak velocity and gain in the anti-saccade task did not change under the hypocapnic hyperventilation condition (Fig. 4*C* and *D*). This stability may reflect the robustness of colliculus and basal ganglia, which integrate excitatory visual and inhibitory cortical inputs to regulate reflexive saccades (Coe & Munoz, 2017; Coe et al., 2019), and therefore maintain anti-saccade performance in response to cerebral hypoperfusion and/or hypocapnia. Importantly, the rate of erroneous pro-saccades also remained unaffected (Fig. 4*A*), further supporting the preservation of inhibitory-control circuits, which share subcortical mechanisms with reflexive saccades driven by visual salience (Goldstein et al., 2022). Similarly, the preserved saccade velocity and gain (Fig. 4*F* and *G*) may imply stable operation of the brainstem burst generator and cerebellar circuits, including the dorsal vermis and fastigial nucleus, which maintain saccade metrics through tightly coupled feedback loops (Barash et al., 1999; Robinson et al., 1993; Shinoda et al., 2019; Takagi et al., 1998) under cerebral hypoperfusion and/or hypocapnia. This interpretation aligns with prior evidence showing that reduced cerebral blood flow had no effect on pro-saccade performance observed in patients with major depressive disorder (Hoffmann et al., 2019). The limited adaptability of saccade gain is also suggested in a previous study wherein the gain remained unchanged even after extensive practice (Dyckman & McDowell, 2005). As for the potential underlying mechanisms, Zani & colleagues (2023) have reported that acute hypoxic breathing increases event-related potential amplitudes in the frontal area but decreases them in the parietal area during visuospatial attention tasks, indicating a possible trade-off in regional activation. Such compensation might have occurred under hypocapnia-induced cerebral hypoperfusion, resulting in the unchanged variables in the anti-saccade task.

## Influence of voluntary hyperventilation on oculomotor control

The observed prolongation of anti-saccade latency under both normocapnic and hypocapnic hyperventilation (Fig. 4*B*) suggests that voluntary hyperventilation can transiently influence goal-directed oculomotor control independently of hypocapnia. One possible explanation is that voluntary modulation of breathing may affect widespread cortical and subcortical networks involved in cortical excitability, emotional processing and interoception (Ashhad et al., 2022), leading to changes in arousal regulation (Buchanan & Janelle, 2021; Song & Lehrer, 2003) and motor performance (Buchanan & Janelle, 2021). Moreover, recent evidence indicates that eye-movement dynamics can vary with the respiratory phase, reflecting transient respiratory–motor coupling (Schaefer et al., 2024; Schaefer et al., 2025). Along these lines, changes in respiratory rhythm can also modulate the excitability of the locus coeruleus–noradrenergic system and other brainstem nuclei projecting to oculomotor and prefrontal regions (Ashhad et al., 2022; Yackle et al., 2017). Such respiration–arousal coupling may transiently induce subtle fluctuations in cortical excitability and/or attentional gating, thereby delaying the suppression of goal-directed anti-saccades. An alternative explanation is that maintaining voluntary rapid breathing imposes an additional cognitive load, as participants must divide attention between respiratory control and task performance, effectively creating a dual-task situation (Rubio et al., 2004), delaying responsiveness of anti-saccade task.

## Perspective

Our findings suggest a link between respiratory activity and oculomotor behaviour (Fig. 5). Accordingly, even transient, non-pathological increases in ventilation – such as those induced by heat stress, hypoxic exposure or psychological stress – may impair visuocognitive performance, potentially elevating the risk of injury and mortality in activities that demand precise visuomotor coordination. Clinically, conditions such as schizophrenia and traumatic brain injury are frequently associated with both hyperventilation and compromised oculomotor control. It is plausible that hyperventilation contributes to impaired oculomotor function in these populations, thereby increasing susceptibility to injury and mortality. To mitigate these potential risks, voluntary regulation of breathing to prevent hypocapnia appears to be an immediate and effective intervention. Alternatively, inhalation of carbon dioxide-enriched air during episodes of hyperventilation may attenuate hypocapnia, thus reducing the above-mentioned risks. Future studies are warranted to directly investigate these possibilities.

## Considerations

End-tidal carbon dioxide partial pressure and middle cerebral artery mean blood velocity decreased ($-0.3 \pm 0.4$ mmHg and $-1.7 \pm 2.1$ cm/s, respectively) over time in the spontaneous breathing condition (i.e., before *versus* during the breathing intervention), indicative of a time-dependent modulation (Fig. 2*B* and *C*). However, these changes were physiologically negligible. In addition, end-tidal carbon dioxide partial pressure increased marginally ($+1.1 \pm 1.5$ mmHg) during normocapnic hyperventilation, likely due to suboptimal tidal volume regulation, possibly because participants were required to focus simultaneously on both the eye movement task and voluntary breathing control. Given that this elevation is very small, we believe its influence such as on ventilatory drive would be negligible; however, this remains unclear in our study.

Despite the slightly elevated end-tidal carbon dioxide partial pressure observed during normocapnic hyperventilation, middle cerebral artery mean blood velocity paradoxically decreased ($-8.5 \pm 8.4$ cm/s; Fig. 2*B* and *C*). Since mean arterial pressure, cardiac responses and percutaneous oxygen saturation, all of which can influence cerebral blood flow, remained unchanged during normocapnic hyperventilation compared to pre-hyperventilation levels, the reason for the observed reduction in middle cerebral artery mean blood velocity remains unclear. In addition, whether, and to what extent, the slight reduction in middle cerebral artery mean blood velocity influenced the latency of anti-saccades remains uncertain. Nonetheless, it is important to note that the other oculomotor variables remained unaffected by normocapnic hyperventilation, indicating that this modest decrease in middle cerebral artery mean blood velocity likely had minimal impact on the overall results.

Unexpectedly, partial pressure of oxygen in the blood analysis was reduced by hypocapnic hyperventilation (Table 3). Given that end-tidal oxygen pressure (Table 1) and percutaneous oxygen saturation (Fig. 2*D*), which reflect arterial oxygen partial pressure and arterial oxygen saturation, respectively, remained unaffected by hypocapnic hyperventilation, we believe arterial oxygen content was not largely unaffected by hypocapnic hyperventilation. It might be that oxygen level in arterialized capillary blood samples from fingertip was influenced by hypocapnia-related factors such as reduction in finger blood flow (Umeda et al., 2008).

We did not directly measure brain activity, brain temperature and reginal differences in brain blood flow. Future studies incorporating neuroimaging techniques (e.g., fMRI, MRS and EEG) would elucidate the underlying neural and physiological mechanisms of hypocapnia-induced alterations in oculomotor behaviour.

## Conclusion

We show that hyperventilation-induced hypocapnia impairs oculomotor function in healthy young adults as evidenced by reduced fixation and saccade frequency, prolonged fixation duration, and shortened scanpath length during free viewing. Additionally, both hypocapnia and hyperventilation *per se* prolong saccadic latency in the anti-saccade task. Based on these findings, voluntary regulation of breathing to prevent hyperventilation or inhalation of carbon dioxide-enriched air to mitigate hypocapnia may serve as effective interventions to reduce the risk of injury and mortality in tasks requiring precise visuomotor coordination.

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

## Additional information

### Data availability statement

All data supporting the results presented here are available from the corresponding author upon reasonable request.

### Competing interests

The authors declare they have no competing interests.

## Author contributions

Y.Y., T.S. and N.F. conceived and designed research. Y.Y. and T.S. performed experiments. Y.Y. and T.S. analysed data; Y.Y., T.S., S.O., T.N. and N.F. interpreted results of experiments. Y.Y. drafted manuscript. Y.Y., T.S., S.O., T.N. and N.F. edited and revised manuscript. All authors have read and approved the final version of this manuscript and agree to be accountable for all aspects of the work in ensuring that questions related to the accuracy or integrity of any part of the work are appropriately investigated and resolved. All persons designated as authors qualify for authorship, and all those who qualify for authorship are listed.

## Funding

No funding was received for this work.

## Acknowledgements

We gratefully acknowledge all research participants and their supporters, whose invaluable contributions made this study possible.

## Keywords

cerebral hypoperfusion, respiratory alkalosis, saccades, visual fixation

## Supporting information

Additional supporting information can be found online in the Supporting Information section at the end of the HTML view of the article. Supporting information files available:

**Peer Review History**

