## [Peer Review History · The Journal of Physiology]

How breathing disrupts vision: hyperventilation-induced hypocapnia impairs oculomotor responses in resting humans

Yusei Yoshimura, Tomoka Sagawa, Seiji Ono, Takeshi Nishiyasu, and Naoto Fujii
DOI: 10.1113/JP289870

Corresponding author(s): Naoto Fujii (fujii.naoto.gb@u.tsukuba.ac.jp)

Review Timeline:

Submission Date:	05-Aug-2025
Editorial Decision:	24-Sep-2025
Revision Received:	30-Nov-2025
Accepted:	02-Jan-2026

Senior Editor: Richard Carson

Reviewing Editor: Frank Powell

Transaction Report:

Dear Dr Fujii,

Re: JP-RP-2025-289870 **"How breathing disrupts vision: hyperventilation-induced hypocapnia impairs oculomotor responses in resting humans"** by Yusei Yoshimura, Tomoka Sagawa, Seiji Ono, Takeshi Nishiyasu, and Naoto Fujii

Thank you for submitting your manuscript to The Journal of Physiology. It has been assessed by a Reviewing Editor and by 2 expert referees and we are pleased to tell you that it is potentially acceptable for publication following satisfactory major revision.

REVISION CHECKLIST:

Please upload two versions of your manuscript text: one with all relevant changes highlighted and one clean version with no

changes tracked. The manuscript file should include all tables and figure legends, but each figure/graph should be uploaded as separate, high-resolution files.

We look forward to receiving your revised submission.

Yours sincerely,

Richard Carson
Senior Editor
The Journal of Physiology

REQUIRED ITEMS

- Author photo and profile. First or joint first authors are asked to provide a short biography (no more than 100 words for one author or 150 words in total for joint first authors) and a portrait photograph. These should be uploaded and clearly labelled together in a Word document with the revised version of the manuscript. See Information for Authors for further details.
- Your manuscript must include a complete Additional Information section, including competing interests; funding; author contributions and acknowledgements.
- Please upload separate high-quality figure files via the submission form.
- Please ensure that any tables are editable and in Word format, and wherever possible, embedded in the article file itself.
- Your paper contains Supporting Information of a type that we no longer publish, including supplementary tables and figures. Any information essential to an understanding of the paper must be included as part of the main manuscript and figures. The only Supporting Information that we publish are video and audio, 3D structures, program codes and large data files. Your revised paper will be returned to you if it does not adhere to our Supporting Information Guidelines.
- A Data Availability Statement is required for all papers reporting original data. This must be in the Additional Information section of the manuscript itself. It must have the paragraph heading 'Data Availability Statement'. All data supporting the results in the paper must be either: in the paper itself; uploaded as Supporting Information for Online Publication; or archived in an appropriate public repository. The statement needs to describe the availability or the absence of shared data. Authors must include in their statement: a link to the repository they have used, or a statement that it is available as Supporting Information; reference the data in the appropriate sections(s) of their manuscript; and cite the data they have shared in the References section. Whenever possible, the scripts and other artefacts used to generate the analyses presented in the paper should also be publicly archived. If sharing data compromises ethical standards or legal requirements then authors are not expected to share it, but must note this in their statement. For more information, see our Statistics Policy.
- Please include an Abstract Figure file, as well as the Figure Legend text within the main article file. The Abstract Figure is a piece of artwork designed to give readers an immediate understanding of the research and should summarise the main conclusions. If possible, the image should be easily 'readable' from left to right or top to bottom. It should show the physiological relevance of the manuscript so readers can assess the importance and content of its findings. Abstract Figures should not merely recapitulate other figures in the manuscript. Please try to keep the diagram as simple as possible and

without superfluous information that may distract from the main conclusion(s). Abstract Figures must be provided by authors no later than the revised manuscript stage and should be uploaded as a separate file during online submission labelled as File Type 'Abstract Figure'. Please also ensure that you include the figure legend in the main article file. All Abstract Figures should be created using BioRender. Authors should use The Journal's premium BioRender account to export high-resolution images. Details on how to use and access the premium account are included as part of this email.

EDITOR COMMENTS

Reviewing Editor:

Your submission has been reviewed by two experts and both find merit in the study but they also raise some significant questions. Notably, there is a question about the value of a "reflexive pro-saccade control" and potential physiological consequences of differences in arterial blood gases between intervention conditions. I am not so concerned about "carry over" effects on buffering and acid-base chemistry raised by one reviewer, given the randomized order of conditions between experiments although that deserves comment. However, I would like to see arterial saturation data and more discussion about any potential effects (or not) predicted for the significant differences in end-tidal CO₂ (Fig 2) and blood gases observed (Tab 5) between conditions and phases on cerebral blood flow and ventilation (as a ventilatory stimulus).

The latter point is of special interest to me because I think some of the value of this study is looking at potential independent effects of increased ventilation on saccades as one measure of neural function. Accordingly, I think the observation of significant effects of hyperventilation on anti-saccade tasks should be included in key points and the discussion could consider alternatives to the cognitive overload possibility. (e.g. Ashhad S, Kam K, Del Negro CA, Feldman JL. Breathing Rhythm and Pattern and Their Influence on Emotion. *Annu Rev Neurosci.* 2022 Jul 8;45:223-247. doi: 10.1146/annurev-neuro-090121-014424. Epub 2022 Mar 8. PMID: 35259917; PMCID: PMC9840384).

Please also see 'Required Items' above.

Senior Editor:

Please note that the Journal does not publish supplementary materials. All relevant information should be provided in the main text.

I would also ask that you consider carefully the comments made by both referees in relation to the statistical analysis design that was employed.

REFeree COMMENTS

Referee #1:

In this groundbreaking study, Yoshimura and colleagues have studied the impact of hypocapnia (reduced carbon dioxide in the blood) to oculomotor responses during a visual search task and an anti-saccade task. To my knowledge, this represents the first attempt to establish connections between physiological mechanisms and cortical control of eye movements. The authors tested the hypothesis that hyperventilation-induced hypocapnia would compromise oculomotor function in resting participants, implementing two control conditions: normal spontaneous breathing and normocapnic hyperventilation. The results demonstrated that hypocapnia specifically reduced fixation and saccade frequency while prolonging fixation duration during visual search. During anti-saccade tasks, both hypocapnic and normocapnic conditions affected saccadic kinematics, with hypocapnia producing more pronounced effects. The authors propose that hypocapnia may impair cortically-mediated eye movement control through reduced cerebral blood flow and/or GABAergic neuronal suppression. The manuscript is well-executed and clearly articulated, though occasionally overly detailed. I offer the following major and minor recommendations for improvement.

Major comments:

- 1) The statistical approach appears unnecessarily complex for this straightforward experimental design. Relegating most analyses to supplementary materials further complicates interpretation. I recommend streamlining the statistical presentation to enhance accessibility and clarity.
- 2) The absence of a reflexive pro-saccade control warrants consideration. A pro-saccade task performed in darkness would provide valuable experimental control, particularly given the authors' cerebral blood flow hypothesis. Since pro-saccades

can be mediated subcortically via the superior colliculus, especially using gap paradigms, they may demonstrate greater resistance to hypocapnic effects. If feasible, incorporating this experiment would strengthen the study's conclusions.

Minor comments:

- 1) Please make sure all the eye-tracking details are reported as per the guidelines proposed in: Dunn, M. J., Alexander, R. G., Amiebenomo, O. M., Arblaster, G., Atan, D., Erichsen, J. T., Ettinger, U., Giardini, M. E., Gilchrist, I. D., Hamilton, R., Hessels, R. S., Hodgins, S., Hooge, I. T. C., Jackson, B. S., Lee, H., Macknik, S. L., Martinez-Conde, S., McIlreavy, L., Muratori, L. M.,...Sprenger, A. (2023). Minimal reporting guideline for research involving eye tracking (2023 edition). *Behavior Research Methods*, 56, 4351-4357. <https://doi.org/10.3758/s13428-023-02187-1>
- 2) Line 279: not clear to me how saccade onset and offset were determined.
- 3) Line 349: "order (three levels...). Not clear which order is being referred to here.
- 4) Figures 2-4 should follow the same consistent design with individual data points shown as well. Figures 3 and 4 do not show individual data points.
- 5) Overall, the Discussion is well-written. I commend the authors for a job well done.

Referee #2:

See the attachment.

END OF COMMENTS

This study tested whether hyperventilation-induced hypocapnia impairs eye movement control in healthy adults. Thirteen participants performed free-viewing and anti-saccade tasks under spontaneous breathing, normocapnic hyperventilation, and hypocapnic hyperventilation. Hypocapnia lowered end-tidal CO₂ and cerebral blood flow, and significantly reduced fixation number, saccades, and scanpath length, while increasing fixation duration during free viewing. In the anti-saccade task, both normocapnic and hypocapnic hyperventilation slowed responses and reduced saccade velocity, with stronger deficits under hypocapnia. These findings show that hyperventilation-mediated hypocapnia disrupts visual fixation and saccadic control.

Although the study is performed with a comprehensive and straightforward methodology, it lacks novelty. It is important to study the impact of hyperventilation on eye movement features. Novel eye movement tasks should be tested instead of free-viewing and anti-saccade, which are commonly used and lack sensitivity in real-world situations. The authors need to elaborate more on choosing the eye movement tasks in the Introduction.

There are major concerns in the methodology of the intervention from the physiological perspective:

1. Did authors desire to do normoxic hypocapnia / normoxic normocapnia ? The inspired oxygen concentration of 16.8% would suggest mild hypoxia (21% is normoxia at sea level). If they indeed tried to do hypoxia then their work is contaminated by the vasodilatation effects of hypoxia, this needs to be explained and clarified (unless the experiment took place at an elevation with ambient pressure and FiO₂ equivalent to 16.8%)?
2. Carry-over effect: Ideally for the measurement of metrics that cause alterations of carbon dioxide levels it would be ideal to provide for much more time for full biochemical recovery, so as to not contaminate the next breathing condition and eye tracking task with the preceding breathing condition. What it means is that hypocapnia inducing breathing for 15 minutes will require in healthy individuals in excess of many hours of full biochemical recovery to normalize tissue buffering capacity.
3. Discussion of effects observed: If #1 and #2 are truly present and not mitigated then it is in my opinion very difficult to disentangle true effects from single interventions versus contamination of carry-over and hypoxia.

The authors have complicated the analysis by adding unnecessary dummy variables or reference levels. It would have been more effective to show the changes in eye movement features before and during the three respective interventions using ANOVA.

Minor Edit:

Line 222: 3.9 +/- 0.4 CO₂ and not O₂.

We thank the editors and reviewers for their thoughtful comments and constructive critiques. We have revised our manuscript based on the feedback provided by both the editors and reviewers. We believe that these revisions have strengthened the overall quality of the paper. Below, we provide a point-by-point response to each comment along with the corresponding revisions. All changes made to the manuscript are highlighted in red.

Reviewing Editor:

Your submission has been reviewed by two experts and both find merit in the study but they also raise some significant questions. Notably, there is a question about the value of a "reflexive pro-saccade control" and potential physiological consequences of differences in arterial blood gases between intervention conditions. I am not so concerned about "carry over" effects on buffering and acid-base chemistry raised by one reviewer, given the randomized order of conditions between experiments although that deserves comment. However, I would like to see arterial saturation data and more discussion about any potential effects (or not) predicted for the significant differences in end-tidal CO₂ (Fig 2) and blood gases observed (Tab 5) between conditions and phases on cerebral blood flow and ventilation (as a ventilatory stimulus).

The latter point is of special interest to me because I think some of the value of this study is looking at potential independent effects of increased ventilation on saccades as one measure of neural function. Accordingly, I think the observation of significant effects of hyperventilation on anti-saccade tasks should be included in key points and the discussion could consider alternatives to the cognitive overload possibility. (e.g. Ashhad S, Kam K, Del Negro CA, Feldman JL. Breathing Rhythm and Pattern and Their Influence on Emotion. *Annu Rev Neurosci.* 2022 Jul 8;45:223-247. doi: 10.1146/annurev-neuro-090121-014424. Epub 2022 Mar 8. PMID: 35259917; PMCID: PMC9840384).

Response:

We thank the reviewing editor for the critical assessment of our manuscript. In response to the reviewers' comments, including the points raised by two reviewers, we have made substantial revisions to the revised manuscript. Please find our point-by-point responses and the corresponding changes.

As for the arterial oxygen saturation data, this had been presented in Table 4 in the initial submission. However, to further highlight this data, we have included this in Figure 2D, clearly indicating that no reductions occurred during any of the interventions. We have also expanded our discussion pertaining to O₂ and CO₂ variables on ventilation and cerebral blood flow (Lines 662-690).

As for the latter point raised by the reviewing editor, we have included statement regarding potential influence of hyperventilation per se on anti-saccade task in the Key points as well as all conclusion statements (Lines 66-68 in the Key points and lines 699-703 in the Conclusion). We have also expanded our discussion on the potential influence of breathing rhythm and pattern on arousal, attention, and emotion (Lines 625-642).

Please also see 'Required Items' above.

Response:

We have checked all instructions and fully addressed.

Senior Editor:

Please note that the Journal does not publish supplementary materials. All relevant information should be provided in the main text.

I would also ask that you consider carefully the comments made by both referees in relation to the statistical analysis design that was employed.

Response:

We have removed supplemental file and moved necessary information to the revised manuscript. We have also addressed the reviewers' concerns, including those related to the statistical analysis. Please see our detailed responses below.

Referee #1:

In this groundbreaking study, Yoshimura and colleagues have studied the impact of hypocapnia (reduced carbon dioxide in the blood) to oculomotor responses during a visual search task and an anti-saccade task. To my knowledge, this represents the first attempt to establish connections between physiological mechanisms and cortical control of eye movements. The authors tested the hypothesis that hyperventilation-induced hypocapnia would compromise oculomotor function in resting participants, implementing two control conditions: normal spontaneous breathing and normocapnic hyperventilation. The results demonstrated that hypocapnia specifically reduced fixation and saccade frequency while prolonging fixation duration during visual search. During anti-saccade tasks, both hypocapnic and normocapnic conditions affected saccadic kinematics, with hypocapnia producing more pronounced effects. The authors propose that hypocapnia may impair cortically-mediated eye movement control through reduced cerebral blood flow and/or GABAergic neuronal suppression. The manuscript is well-executed and

clearly articulated, though occasionally overly detailed. I offer the following major and minor recommendations for improvement.

Major comments:

1) The statistical approach appears unnecessarily complex for this straightforward experimental design. Relegating most analyses to supplementary materials further complicates interpretation. I recommend streamlining the statistical presentation to enhance accessibility and clarity.

Response:

We have simplified the analytical framework to enhance clarity and interpretability while maintaining statistical rigor. Specifically, we now use a single generalized or linear mixed model for each eye movement variable, defined as:

$$Y \sim (\text{intercept}) + \text{intervention} + \text{phase} + \text{order} + \text{intervention} \times \text{phase} \\ + (\text{intercept} \mid \text{participant})$$

Here, intervention (three levels: spontaneous breathing, hypocapnic hyperventilation, normocapnic hyperventilation) and phase (two levels: before vs. during breathing intervention) were included as fixed effects. In response to reviewer #2 regarding potential carry-over effects, we additionally included order (the implementation order of the three intervention conditions; three levels: first, second, third) as a covariate in the fixed effects. Participants were modeled as random intercepts to account for interindividual variability, as evident from the baseline differences observed across individual data points in Figures 3 and 4. This simplified model structure avoids overparameterization while adequately capturing the essential experimental factors and sources of variability. We believe this revision substantially improves accessibility and transparency of the statistical analysis. Please see revised Methods and Results section (Lines 397-416 in the Methods and lines 455-491 in the Results).

2) The absence of a reflexive pro-saccade control warrants consideration. A pro-saccade task performed in darkness would provide valuable experimental control, particularly given the authors' cerebral blood flow hypothesis. Since pro-saccades can be mediated subcortically via the superior colliculus, especially using gap paradigms, they may demonstrate greater resistance to hypocapnic effects. If feasible, incorporating this experiment would strengthen the study's conclusions.

Response:

We fully agree that inclusion of a reflexive pro-saccade control would provide a valuable experimental reference. To address this important point, we have included peak velocity and

gain of erroneous pro-saccades data as presented in the revised manuscript (Lines 332-334 in the Methods and Fig. 4E–G in the Results). These additional analyses consistently showed that hypocapnic hyperventilation did not affect these variables, supporting our interpretation that reflexive, subcortically mediated saccadic control remained functionally intact under hypocapnic conditions. Furthermore, we have expanded the Discussion section to explicitly address this aspect (Lines 603-623). Importantly, the erroneous pro-saccades observed during the anti-saccade task serve as a valid surrogate index of reflexive oculomotor control based on previous research suggesting that erroneous pro-saccades during the anti-saccade task share neural substrates with reflexive pro-saccades, primarily mediated by the superior colliculus (Goldstein et al., 2022; doi: <https://doi.org/10.7554/eLife.76964>). Based on this, we believe the aforementioned additional analyses adequately addressed the reviewer's concern without having additional experiment.

Minor comments:

1) Please make sure all the eye-tracking details are reported as per the guidelines proposed in: Dunn, M. J., Alexander, R. G., Amiebenomo, O. M., Arblaster, G., Atan, D., Erichsen, J. T., Ettinger, U., Giardini, M. E., Gilchrist, I. D., Hamilton, R., Hessels, R. S., Hodgins, S., Hooge, I. T. C., Jackson, B. S., Lee, H., Macknik, S. L., Martinez-Conde, S., McIlreavy, L., Muratori, L. M.,...Sprenger, A. (2023). Minimal reporting guideline for research involving eye tracking (2023 edition). *Behavior Research Methods*, 56, 4351-4357. <https://doi.org/10.3758/s13428-023-02187-1>

Response:

We have thoroughly revised and expanded the Methods section to ensure including necessary information as per the guideline. Specifically, the following information has been added or clarified in the revised manuscript (Signal latencies: lines 248-249; Software and firmware versions: lines 285-289; Measurement uncertainty: lines 289-291; Calibration: lines 294-299; Data loss: lines 324-326 and 334-336). These additions ensure compliance with the Minimal reporting guideline for research involving eye tracking and enhance the methodological transparency and reproducibility of the study.

2) Line 279: not clear to me how saccade onset and offset were determined.

Response:

We have expanded the description of how saccade onset and offset were determined, following the velocity-based detection algorithm proposed by Nyström & Holmqvist (2010) (Lines 308-318).

3) Line 349: "order (three levels...). Not clear which order is being referred to here.

Response:

The text has been revised for clarity (Lines 406-408).

4) Figures 2-4 should follow the same consistent design with individual data points shown as well. Figures 3 and 4 do not show individual data points.

Response:

We have revised Figures 3 and 4 by including individual data points.

5) Overall, the Discussion is well-written. I commend the authors for a job well done.

Response:

Thank you for the positive comment.

Referee #2:

This study tested whether hyperventilation-induced hypocapnia impairs eye movement control in healthy adults. Thirteen participants performed free-viewing and anti-saccade tasks under spontaneous breathing, normocapnic hyperventilation, and hypocapnic hyperventilation. Hypocapnia lowered end-tidal CO₂ and cerebral blood flow, and significantly reduced fixation number, saccades, and scanpath length, while increasing fixation duration during free viewing. In the anti-saccade task, both normocapnic and hypocapnic hyperventilation slowed responses and reduced saccade velocity, with stronger deficits under hypocapnia. These findings show that hyperventilation-mediated hypocapnia disrupts visual fixation and saccadic control. Although the study is performed with a comprehensive and straightforward methodology, it lacks novelty. It is important to study the impact of hyperventilation on eye movement features. Novel eye movement tasks should be tested instead of freeviewing and anti-saccade, which are commonly used and lack sensitivity in real-world situations. The authors need to elaborate more on choosing the eye movement tasks in the Introduction.

Response:

We appreciate the reviewer's insightful comment regarding task selection and novelty, which we have now clarified in the revised Introduction. Specifically, we emphasize that the use of these well-established paradigms—the free-viewing and anti-saccade tasks—was a deliberate methodological choice to ensure both experimental rigor and interpretability. These paradigms have been extensively validated across cognitive neuroscience, neurophysiology, and clinical research, providing a solid foundation for quantifying distinct yet complementary aspects of oculomotor control. The free-viewing task captures spontaneous fixation and saccadic dynamics that reflect natural visual exploration and attentional allocation, whereas the anti-saccade task

isolates top-down inhibitory and executive processes mediated by well-characterized frontal and subcortical networks. By employing both tasks, we were able to examine how hyperventilation-induced hypocapnia influences both automatic and volitional components of eye movement control within a robust and neurophysiologically interpretable framework. Furthermore, using canonical tasks facilitates direct comparison with a large body of previous literature, allowing our findings to serve as a reproducible baseline for future studies employing more complex or ecologically valid oculomotor paradigms. These clarifications have been incorporated into the Introduction (Lines 134-147).

There are major concerns in the methodology of the intervention from the physiological perspective:

1. Did authors desire to do normoxic hypocapnia / normoxic normocapnia ? The inspired oxygen concentration of 16.8% would suggest mild hypoxia (21% is normoxia at sea level). If they indeed tried to do hypoxia then their work is contaminated by the vasodilatation effects of hypoxia, this needs to be explained and clarified (unless the experiment took place at an elevation with ambient pressure and FiO₂ equivalent to 16.8%)?

Response:

The lower inspiratory O₂ concentrations ($16.8 \pm 0.4\%$ during the hypocapnic condition and $16.5 \pm 0.3\%$ during the normocapnic condition) were intentionally set to maintain normoxic conditions, as hyperventilation can slightly increase arterial O₂ partial pressure. This explanation has been added to the Methods section for clarity (Lines 241-243). Please note that percutaneous oxygen saturation remained above 95% under all conditions (Fig. 2D), clearly indicating that participants were normoxic (Lines 441-442).

2. Carry-over effect: Ideally for the measurement of metrics that cause alterations of carbon dioxide levels it would be ideal to provide for much more time for full biochemical recovery, so as to not contaminate the next breathing condition and eye tracking task with the preceding breathing condition. What it means is that hypocapnia inducing breathing for 15 minutes will require in healthy individuals in excess of many hours of full biochemical recovery to normalize tissue buffering capacity.

Response:

To address reviewers' concern, we included "order" (the sequence of the three intervention conditions) as a covariate in generalized linear mixed models (GLMM) and linear mixed models (LMM) (Lines 397-411). Therefore, we believe that we were able to isolate the influence of

hypocapnia independently of carry-over effect. Moreover, the order of the three breathing interventions was randomized across participants to minimize potential order effects.

3. Discussion of effects observed: If #1 and #2 are truly present and not mitigated then it is in my opinion very difficult to disentangle true effects from single interventions versus contamination of carry-over and hypoxia.

Response:

As described above, we feel we were able to well control potential carry-over effect.

The authors have complicated the analysis by adding unnecessary dummy variables or reference levels. It would have been more effective to show the changes in eye movement features before and during the three respective interventions using ANOVA.

Response:

We agree that our initial model description may have appeared unnecessarily complex. To address this concern, we have simplified our modeling strategy by consolidating it into a single mixed-effects model structure. Nevertheless, we believe that the use of linear mixed models (LMMs) and generalized linear mixed models (GLMMs) remains the most appropriate analytical approach for oculomotor responses, rather than a repeated-measures ANOVA. Mixed models offer two key advantages relevant to our design:

(1) they allow inclusion of participant as a random intercept, thereby accounting for interindividual variability and isolating the effect of hypocapnic hyperventilation more accurately. In this study, eye movement parameters exhibited considerable interindividual variability—for instance, scanpath length during the free-viewing task ranged from ~30 in some participants to ~150 in others. The intraclass correlation coefficient (ICC) for scanpath length was 0.46, indicating that 46% of total variance was attributable to between-subject differences. Such variability could obscure the underlying effects of hypocapnia if not modeled appropriately; thus, mixed-effects modeling is essential for accurately characterizing oculomotor responses.

(2) they enable adjustment for order effects (carry-over, fatigue, or practice), which were a concern raised by your comment, by including “order” as a fixed covariate.

Regarding the reviewer’s comment on the inclusion of dummy variables and reference levels, we appreciate this constructive suggestion. Upon careful consideration, we recognized that these model specifications were unnecessary for the clarity of presentation. Accordingly, we have removed these descriptions from the text and reanalyzed the data to ensure consistency. This revision simplifies the statistical modeling without altering the main results and substantially

improves the transparency and readability of the Methods and Results sections (Lines 397-416 in the Methods and lines 455-491 in the Results).

Minor Edit:

Line 222: 3.9 +/- 0.4 CO₂ and not O₂.

Response:

We have corrected the error.

Dear Dr Fujii,

Re: JP-RP-2025-289870R1 "**How breathing disrupts vision: hyperventilation-induced hypocapnia impairs oculomotor responses in resting humans**" by Yusei Yoshimura, Tomoka Sagawa, Seiji Ono, Takeshi Nishiyasu, and Naoto Fujii

We are pleased to tell you that your paper has been accepted for publication in The Journal of Physiology.

Yours sincerely,

Richard Carson
Senior Editor
The Journal of Physiology

IMPORTANT POINTS TO NOTE FOLLOWING ACCEPTANCE OF YOUR PAPER:

- **IMPORTANT NOTICE ABOUT OPEN ACCESS:** To assist authors whose funding agencies mandate immediate public access to published research findings, The Journal of Physiology allows authors to pay an Open Access (OA) fee to have their papers made freely available immediately on publication.

- You can help your research get the attention it deserves! Check out Wiley's free Promotion Guide for best-practice recommendations for promoting your work at: www.wileyauthors.com/eeo/guide. You can learn more about Wiley Editing Services which offers professional video, design, and writing services to create shareable video abstracts, infographics, conference posters, lay summaries, and research news stories for your research at: www.wileyauthors.com/eeo/promotion.

- If you would like to receive our 'Research Roundup', a monthly newsletter highlighting the cutting-edge research published in The Physiological Society's family of journals (The Journal of Physiology, Experimental Physiology, Physiological Reports, The Journal of Nutritional Physiology and The Journal of Precision Medicine: Health and Disease), please click this link, fill in your name and email address and select 'Research Roundup':
<https://www.physoc.org/journals-and-media/membernews>

EDITOR COMMENTS

Reviewing Editor:

Thank you for this responsive revision that seems to have satisfied the major concerns expressed by the expert reviewers. The work answers a significant physiological question (effects of hypocapnia versus hyperventilation per se) about oculomotor control, and raises some interesting questions for future study.

REFeree COMMENTS

Referee #1:

While the manuscript has improved considerably, particularly following the authors' revisions to the pro-saccade metrics, I believe there are a couple of major issues in the Results and Discussion sections that require further attention.

Major Comments:

1) Figure 4: Pro-saccade results are presented very cursorily. I think the erroneous pro-saccade results should be presented separately for clarity and not buried in the same figure as the anti-saccade results.

2) There is an effect of intervention on two of the pro-saccade metrics (velocity and gain). These are the important metrics. I recommend the authors run proper statistical tests and post-hocs to ascertain which specific condition(s) may be causing these differences.

3) That would also require re-interpreting the results. The latencies of pro-saccades are much shorter than anti-saccades. This suggests reconsidering the mechanisms by which these pro-saccade metrics are being impacted.

4) Figure 5: This is too simplistic, given the combined results on anti-saccades and pro-saccades. Your model should be able to explain both the results and generate testable predictions for both cortical and subcortical involvement of saccade generation. I recommend reviewing Figure 2 of Coe and Munoz (2017) and reconciling your results and model with what is known about saccades and anti-saccades in the context of the physiological interventions you implemented.

Coe, B. C., & Munoz, D. P. (2017). Mechanisms of saccade suppression revealed in the anti-saccade task. *Philosophical Transactions of the Royal Society B: Biological Sciences*, 372(1718), 20160192. <https://doi.org/10.1098/rstb.2016.0192>

Minor Comments:

5) Thank you for making the changes based on the recommendations of Dunn et al. Please cite the paper as well so readers know the basis for providing these details. For full disclosure, this Reviewer is an author on that paper.

I appreciate the authors' careful attention to these revisions and look forward to seeing the updated manuscript.

Referee #2:

The authors have addressed all the comments by revising the manuscript. The paper's readability has improved.